# An Approach of Improving the Efficiency of Software Fault Localization based on Feedback Ranking Information

**Bo Yang** [1,2,3,†], **Xiaowen Ma** [1], **Haoran Guo** [3], **Yuze He** [3] **and Fu Xu** [1,2,*]

1    School of Information Science and Technology, Beijing Forestry University, Beijing 100083, China
2    Engineering Research Center for Forestry Oriented Intelligent Information Processing, National Forestry and Grassland Administration, Beijing 100083, China
3    School of Information Science and Technology, North China University of Technology, Beijing 100144, China
*    Correspondence: xufu@bjfu.edu.cn
†    These authors contributed equally to this work.

**Abstract:** Fault localization, a critical process of software debugging, can be implemented by ranking program statements according to their suspiciousness of being faulty, which, in turn, is calculated based on the execution behaviors of test cases. The performance of fault localization will deteriorate if the actual faulty statement is ranked low in suspiciousness. Intuitively speaking, the quality of the used test cases affects the suspiciousness ranking and thus the efficacy of fault localization. As such, it is necessary to generate test cases with "better" quality to improve the chance of faulty statements being ranked as highly suspicious. In this paper, we propose a software fault localization approach based on feedback ranking information, namely FLFR, according to an improved genetic algorithm. The starting point of the new method is the execution of a set of test cases, which gives a preliminary suspiciousness ranking of program statements. The improved genetic algorithm is iteratively applied to generate new test cases. The new method is evaluated through a series of experiments on four C programs and two Java programs. The experimental results show that the test cases automatically generated by the method can improve the suspiciousness ranking of the faulty statement, and thus enhance the performance of fault localization.

**Keywords:** fault localization; test case generation; genetic algorithm; fitness function; random test

Software testing company Tricentis gave a report that revealed that software faults caused USD 1.7 trillion in financial losses (https://www.techrepublic.com/article/report-software-failure-caused-1-7-trillion-in-financial-losses-in-2017/, accessed on 1 January 2023). Software systems are increasing in size and complexity. If there is a fault in the software, it will bring a lot of trouble to users, developers, and maintainers. It can sometimes lead to serious and even fatal safety problem.

To find faults in software more quickly, many researchers have carried out studies on software fault localization. One of the typical methods is the Spectrum-based fault localization (SBFL) [1–3]. The program spectrum contains the coverage information of the program in the process of running the test case. The elements of the program can be statements, statement blocks, predicates, methods, variables, etc. A program spectrum is a representation used to describe program behavior. It has been widely used in program understanding, regression testing, and other fields. SBFL divides the program into basic units (functions, statements, predicates, definition-use chains, etc.) according to the selected program spectrum. We can evaluate the degree of correlation between each program unit and program failure by the coverage information of the program unit obtained after executing the test cases.

SBFL techniques are widely studied. For example, Jones et al. [1] presented a technique that recorded the statement coverage status. For each statement in the code, the execution status of the failed test case and the pass test case is recorded, which can help the user locate potentially faulty statements. Wong et al. [4] applied a crosstab-based statistical

method, which can obtain the coverage information of each executable statement, and the execution result (pass or fail). Ghandehari et al. [5] presented a technique that obtained a ranking of statements in terms of their likelihood of being faulty by leveraging the result of combinatorial testing. Due to its effectiveness, SBFL has been used by developers for debugging in practice [6,7].

In addition, there are a few software fault localization studies with deep learning techniques. For example, Zhang et al. [8] proposed a convolutional neural network-based fault localization method CNNFL (Convolutional Neural Network Fault localization).

However, the performance of fault localization will be deteriorated if the actual faulty statement is ranked low in suspiciousness. Intuitively speaking, the quality of the used test cases affects the suspiciousness ranking and thus the efficacy of fault localization. As such, it is necessary to generate test cases with "better" quality to improve the chance of faulty statements being ranked as highly suspicious. To this end, some researchers have improved the efficiency of SBFL from the perspective of identifying coincidental correctness (CC) test cases [9–13].

Coincidental correct test cases have been investigated with negative effects on coverage-based fault localization, which implement the faulty statement but with correct output. Assi et al. [13] conducted an empirical study on the impact of coincidental test cases on three tasks, namely test case reduction, test case prioritization, and SBFL. Results show that when all CC test cases were deleted, the effect of fault localization is improved. In addition, they mentioned in the threat analysis that clearing all coincidental test cases in multi-error situations reduces the effect of error localization.

Existing methods often rely heavily on the quality and comprehensiveness of test cases. If the test cases are not comprehensive or do not cover all possible scenarios, the method may fail to locate the fault. In addition, many methods are based on statistical analysis and lack a deep understanding of the context and semantics of the software. This often leads to misinterpretations of the fault's causes and effects.

Considering that the effectiveness of the SBFL technique is closely related to the test cases, it may have a positive effect from the perspective of increasing the test cases. To this end, we propose a software fault localization approach based on feedback ranking information (FLFR), which can help to generate test cases based on fault localization intermediate result information. The goal is to increase the rank of the statement containing the fault in the set of suspected faults. FLFR is designed to be highly accurate in identifying the source of software faults. They utilize user feedback and historical data to narrow down the possible location of faults. FLFR learn from previous faults and solutions, improving their effectiveness over time. This continuous learning approach helps to prevent the recurrence of similar faults in the future.

The method applies a set of test cases to execute the program under test. Then, we can obtain the program spectrum information after execution and convert the program spectrum information into coverage matrices and result vectors. Next, we analyze the 13 typical spectrum-based methods and obtain the ranking of the suspected faults of the statements contained in the program under test. These rankings will be used to improve the fitness function construction of the genetic algorithm (GA). We use an improved genetic algorithm to generate new test cases. The process is iterative. At last, we get the top-ranked statements that actually contain faults after multiple iterations.

The contribution of the paper is three-fold, as summarized in the following:

- We propose FLFR, which improves the genetic algorithm. Specifically, we add the localization parameter of the suspected faulty sentence to the genetic algorithm's fitness function. This way, when generating new test cases, we can improve the coverage of these suspected fault statements. In addition, we have adaptively designed the genetic algorithm's crossover probability and mutation operator. On the one hand, the diversity of the population is considered, and, on the other hand, the overall search efficiency is improved.

- We use FLFR in 13 typical fault spectrum calculation methods. We analyzed the effects of the FLFR method from the perspective of different failure types and different iterations.
- We conduct experiments on six datasets including C and Java. In addition, we compare 13 typical software fault localization approaches. The experiments show that the test cases supplemented by FLFR can improve the efficiency of software fault localization.

The rest of this paper is organized as follows. Section 1 presents the research motivation. The approach framework and improved GA algorithm are described in Section 2. Section 3 presents the experiments for evaluating FLFR, the results of which are reported in Section 4. Section 5 presents the related work. Finally, we conclude this paper in Section 6.

## 1. Motivation

### 1.1. Related Notations

Software fault localization contains some concepts, including fault location based on the program spectrum, coverage matrix, etc. To clearly illustrate our proposed approach, we will describe these concepts.

- Program spectrum-based fault localization: The program spectrum-based method is an efficient fault localization method. The program spectrum refers to a collection of runtime code coverage information. Jones et al. [1] proposed four factors: $N_{CF}$: the number of times the statement is covered by the fail test case, $N_{UF}$: the number of times the statement is not covered by the fail test case, $N_{CS}$: the number of times the statement is covered by the pass test case, and $N_{US}$: the number of times the statement is not covered by the pass test case. Jones et al. uses these four factors to construct a formula to calculate the suspiciousness of each statement in the program and then generates a list of suspicious rankings.
- Coverage matrix: Suppose a program $P$ has a total of $M$ statements, and the program $P$ contains single or multiple faults. Using $N$ test cases to execute the program $P$, the execution set of the program $P$ based on the test cases will be obtained. This set will form a coverage matrix. The behavior of the matrix is the coverage of all statements when each test case is executed. If a statement is covered once during execution, the corresponding value in the matrix is incremented by 1. If a statement is not covered during execution, the corresponding value in the matrix is 0. The columns of the matrix are the set of all statements in program $P$.
- Result vector: The result vector refers to the test judgment value of the executed test case. The value of the test result includes 1 and 0. The value of 1 stands for the failed test case, and the value of 0 stands for the successful test case.
- Test Case Matrix: The test case matrix refers to a square matrix whose dimension is equal to the number of statements in the program $P$. The diagonal elements of the square matrix are all 1, and the remaining elements are all 0.

### 1.2. A Motivation Example

Table 1 shows the median calculation program. The fault localization is at S6, and m equals b should be m equals a. T1 and T2 are 2 test cases, and the specific value is (4, 2, 5) and (2, 2, 3). We mark the execution times of each line of the program. After the test cases are executed, the test oracle of a successful test case is pass, otherwise, the test oracle is fail. The spectral information for this program is thus obtained. We use ochiai1 [14] to calculate the failure suspect degree of the statement. The results show that the suspiciousness score (Sus_score) of S6 is the first, but it ranks fifth because of the order.

**Table 1.** A motivation example.

| Statement | Program | Coverage | | $N_CF$ | $N_CS$ | $N_UF$ | $Sus_score$ (Ochiai1) | Ranking |
|---|---|---|---|---|---|---|---|---|
| | | T1 | T2 | | | | | |
| | void mid(int a,int b,int c) | | | | | | | |
| S1 | {m=c; | 1 | 1 | 1 | 1 | 0 | 0.707 | 1 |
| S2 | if(b<c){ | 1 | 1 | 1 | 1 | 0 | 0.707 | 2 |
| S3 | if(a<b){ | 1 | 1 | 1 | 1 | 0 | 0.707 | 3 |
| S4 | m=b;} | 0 | 0 | 0 | 0 | 1 | 0 | 7 |
| S5 | else if(a<c){ | 1 | 1 | 1 | 1 | 0 | 0.707 | 4 |
| S6 | m=b;}} | 1 | 1 | 1 | 1 | 0 | 0.707 | 5 |
| S7 | else{ | 0 | 0 | 0 | 0 | 1 | 0 | 8 |
| S8 | if(a>b){ | 0 | 0 | 0 | 0 | 1 | 0 | 9 |
| S9 | m=b;} | 0 | 0 | 0 | 0 | 1 | 0 | 10 |
| S10 | else if(a>c){ | 0 | 0 | 0 | 0 | 1 | 0 | 11 |
| S11 | m=a;}} | 0 | 0 | 0 | 0 | 1 | 0 | 12 |
| S12 | print(m); | 1 | 1 | 1 | 1 | 0 | 0.707 | 6 |
| | Test Oracle | Fail | Pass | | | | | |

In general, statements with the highest suspicion scores are checked by the programmer first. However, the current results may not be perfect. To make the list of suspiciousness scores more helpful for programmers to fault localization quickly, we consider adding test cases that can increase the coverage of statements with the highest suspiciousness scores. The reason for this is to make the results of the suspiciousness score more reasonable by adding more test cases.

The new test case T3 (4, 2, 3), T4 (5, 7,8), T5 (6, 5, 8), and T6 (9, 5, 2) is shown in Table 2. We continue to use the ochiai1 fault localization method. The result shows that S6 is ranked 1.

**Table 2.** Example of illustrating fault localization by adding test cases.

| Statement | Program | Coverage | | | | | | $N_CF$ | $N_CS$ | $N_UF$ | $Sus_score$ (Ochiai1) | Ranking |
|---|---|---|---|---|---|---|---|---|---|---|---|---|
| | | T1 | T2 | T3 | T4 | T5 | T6 | | | | | |
| | void mid(int a,int b,int c) | | | | | | | | | | | |
| S1 | {m=c; | 1 | 1 | 1 | 1 | 1 | 1 | 2 | 4 | 0 | 0.577 | 4 |
| S2 | if(b<c){ | 1 | 1 | 1 | 1 | 1 | 1 | 2 | 4 | 0 | 0.577 | 5 |
| S3 | if(a<b){ | 1 | 1 | 1 | 1 | 1 | 0 | 2 | 3 | 0 | 0.632 | 2 |
| S4 | m=b;} | 0 | 0 | 0 | 0 | 0 | 0 | 0 | 0 | 2 | 0 | 7 |
| S5 | else if(a<c){ | 1 | 1 | 1 | 1 | 1 | 0 | 0 | 0 | 2 | 0.632 | 3 |
| S6 | m=b;}} | 1 | 1 | 0 | 0 | 1 | 0 | 2 | 3 | 0 | 0.816 | 1 |
| S7 | else{ | 0 | 0 | 0 | 0 | 0 | 1 | 2 | 1 | 0 | 0 | 8 |
| S8 | if(a>b){ | 0 | 0 | 0 | 0 | 0 | 1 | 0 | 1 | 2 | 0 | 9 |
| S9 | m=b;} | 0 | 0 | 0 | 0 | 0 | 1 | 0 | 1 | 2 | 0 | 10 |
| S10 | else if(a>c){ | 0 | 0 | 0 | 0 | 0 | 0 | 0 | 1 | 2 | 0 | 11 |
| S11 | m=a;}} | 0 | 0 | 0 | 0 | 0 | 0 | 0 | 1 | 2 | 0 | 12 |
| S12 | print(m); | 1 | 1 | 1 | 1 | 1 | 1 | 2 | 4 | 0 | 0.577 | 6 |
| | Test Oracle | Fail | Pass | Pass | Pass | Pass | Fail | | | | | |

As seen from this example, when we use an existing test case, at least five statements need to be checked to locate the faulty statement in line 6. However, we add three test cases to locate the faulty statement in line 6, and only one statement needs to be checked at the fastest.

In addition, we found that the addition of test cases helped find statements containing failures more quickly because they significantly increased the number of times statements were covered in execution success and execution failure.

This gives us some inspiration, if we can increase the number of test cases, so that the number of statements covered by successful execution and error execution increases, it will be possible to help us troubleshoot more efficiently.

## 2. Methodology

### 2.1. The Framework of FLFR

The framework of the method proposed in this paper is shown in Figure 1. Firstly, we use test cases to execute the program, which have been instrumented in advance. After executing the program under test by using the test cases, we will get the coverage information of each statement.

We repeatedly utilize the information of software fault localization, specifically, it refers to the list of likelihood scores for each fault localization, which provides information on potential faults and is very important for generating new test cases. However, with the existing technology based on genetic algorithm, it is difficult to use the information of software fault localization. To this end, we improved the genetic algorithm by adding the position parameter of the suspected fault statement to its severity function. In this way, when generating new test cases, we can maximize the coverage of these suspected fault statements.

The process of FLFR is as follows. Firstly, we input the test case set and the program under test. After executing the program under test, we can obtain the statement coverage information of the program under test during the execution process, which can be used to construct a coverage matrix. We use the software fault location method to obtain a list of suspected fault statements. Secondly, we need to judge whether iteration is required. The iteration condition is limited by the max number of iterations. If the current number of iterations is less than the max number of iterations, we need to use the improved genetic algorithm for generating new test cases. The new test cases will be added to the original test cases. Then, we need to do the whole process again. If the current number of iterations is greater than or equal to the max number of iterations, we need to terminate the iterative process. At last, we obtain the final list of suspected fault statements.

We use iterative fault location because the result information of each fault location multiple times is useful. If we can increase the ranking of the reported failure statements every iteration, then we can make multiple iterations in this direction. Then the final positioning result will definitely be better than the first positioning result. The genetic algorithm is able to increase the number of times a statement is covered when execution succeeds and execution fails, based on our proposed fitness function. In this way, according to the findings in the motivation example, we will likely get good fault location results.

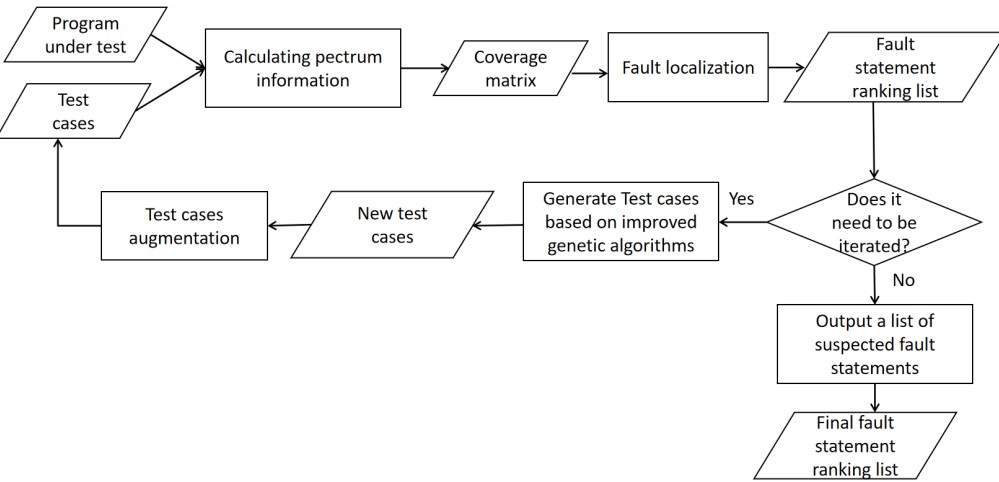

**Figure 1.** The framework of FLFR.

### 2.2. Suspected Fault Statement Ranking Calculation

The fault localization method based on program spectrum statistics generally uses the program spectrum obtained by the execution of test cases. We can obtain the suspiciousness score of each line of the program by using the existing suspiciousness score calculation

formula to calculate the proportion of successful test cases and failed test cases, respectively. Then, we can get a ranking list according to the suspiciousness score to display the code statements that may fail. We use 10 fault localization methods based on spectral statistics in our framework, which are tarantula, Kuczynski, Jaccard, ochiai1, ochiai2, d2star, d3star, op2, gp02, and gp19. In addition, there are three neural network methods, which are BPNN, RBFNN, and CNNFL.

### 2.3. Improved Genetic Algorithm for Fault Localization

2.3.1. Improved Genetic Algorithm

A genetic algorithm is a typical heuristic search algorithm that simulates and abstracts the biological genetic and evolutionary processes in nature and uses key steps such as selection, crossover, and mutation to complete the search for the optimal solution to a specific problem. This paper improves the genetic algorithm by adding the position parameter of potential faulty statements generated by the spectrum-based methods to the fitness function, so as to improve the coverage of faulty statements.

In addition, this paper also designs an adaptive crossover and mutation operator to dynamically adjust the crossover probability and mutation probability by measuring the variety of the population, thereby improving the overall search efficiency. An elitist preservation strategy is adopted for good individuals. The genes of selected good individuals will not be modified by crossover and mutation operations and directly enter the next generation of the population.

The general steps of our improved genetic algorithm are shown in Algorithm 1. The inputs of the algorithm are Initial population P which contains the initial test cases. $P_{c_{basic}}$ is the basic crossover probability. $P_{c_{variance}}$ is the variance of crossover probability. $P_{m_{basic}}$ is the basic mutation probability. $P_{m_{variance}}$ is the variance of mutation probability. max_round is the preset maximum number of iterations.

---

**Algorithm 1:** Improved Genetic Algorithm

---

**Input:** $P$, $P_{c_{basic}}$, $P_{c_{variance}}$, $P_{m_{basic}}$, $P_{m_{variance}}$, stop_condition, max_round
**Output:** New population $P'$
round = 0;
$P' = P$
**while** *round < max_round **and** not stop_condition* **do**
    **for** *individual in $P'$* **do**
        calculate fit(individual);
    calculate variety $V$;
    update $P_{c_{variance}}$, $P_{m_{variance}}$ based on the $V$
    update $P_c$ , $P_m$;
    create roulette;
    select parents using roulette;
    $P'$+= crossover (parents,$P_c$);
    **for** *individual in $P'$* **do**
        $P'$ += mutate (individual, $P_m$)
    Sift ($P'$);
    round++;
**return** $P'$

---

2.3.2. Fitness Function

We assume that the list of suspicion scores obtained for each fault localization is credible, and the top-ranked sentences have a higher probability of containing faults, and we consider these top-ranked sentences to be key sentences. To this end, we design a fitness function based on key sentences.

Here, $t$ represents the test case, the *count* represents the total number of lines of the program under test, program $S$ represents the set composed of the line numbers of the first $n$ suspected faults after the fault localization is calculated and ranked. $cov(t, Si)$ represents whether test case t covers the $i$-th suspicious fault row. The value is 1 if covered, and 0 if not covered.

We use this fitness function calculation to calculate the fitness for each test case. Such a fitness function depends on the result of fault localization. The test case generation of this round is ranked according to the fault localization result of the previous round, and more test cases are generated to override the program statement with the highest failure ranking. Since the fault localization results of each round are different, the generation strategy of test cases in the iterative process will also be adjusted accordingly, which is conducive to the development of fault localization results in a better direction.

$$fit(t) = \begin{cases} \frac{\sum_{i=0}^{n} cov(t,S_i)}{count} & \text{(if t is fail)} \\ 0 & \text{(if t is pass)} \end{cases} \tag{1}$$

### 2.3.3. Adaptive Genetic Operator

Population diversity is the embodiment of the wide distribution of individual genes in the population, which is an aspect that the population needs to ensure in the process of overall evolution.

Crossover is a very important part of the genetic algorithm. It replaces the structure of two-parent parts to generate a new individual. The crossover probability PC is the key parameter, which can affect the evolution of the population to a certain extent and ultimately the searchability of the genetic algorithm. This paper adopts the adaptive crossover probability, which mainly hopes to increase or save the crossover probability in the iterative process according to the diversity of the parent population.

The value process of adaptive crossover probability is as follows. When the fitness function value of individuals in the current population is less than that of the previous generation population, it will be considered to increase the crossover probability value, specifically, we use the basic crossover probability value plus the value of probability change. When the fitness function value of the individual in the current population is greater than or equal to the previous generation population, the crossover probability value is not changed.

The calculation formula of adaptive crossover probability is as follows:

$$Pc_i = \begin{cases} Pc_{\text{basic}} & V_i \geqslant V_{i-1} \\ Pc_{\text{basic}} + Pc_{\text{variance}} \; \text{sigmoid}\left(\frac{V_{i-1}}{V_i}\right) & V_i < V_{i-1} \end{cases} \tag{2}$$

$Pc_{\text{basic}}$ refers to the preset basic crossover probability, and $Pc_{\text{variance}}$ refers to the preset probability change. The sigmoid function is used to calculate the numerical normalization. $V_i$ refers to the fitness function value of the individual of the current population, and $V_{i-1}$ refers to the fitness function value of the individual of the previous generation.

A mutation is an auxiliary way of population evolution in a genetic algorithm. Changing mutation operators can affect population evolution to a certain extent and also affect the searchability of a genetic algorithm. This paper uses an adaptive mutation operator, which mainly hopes to increase or save the mutation probability in the iterative process according to the diversity of the parent population.

The value process of adaptive mutation probability is as follows. When the fitness function value of individuals in the current population is less than that of the previous generation population, it will be considered to increase the mutation probability value. Specifically, we use the basic mutation probability value plus the value of probability change. When the fitness function value of the individual in the current population is

greater than or equal to the previous generation population, the variation probability value is not changed.

The variation probability can be calculated by the following formula:

$$Pm_i = \begin{cases} Pm_{\text{basic}} & V_i \geqslant V_{i-1} \\ Pm_{\text{basic}} + Pm_{\text{variance}} \text{ sigmoid}\left(\dfrac{V_{i-1}}{V_i}\right) & V_i < V_{i-1} \end{cases} \tag{3}$$

where $Pm_{\text{basic}}$ refers to the preset basic variation rate. $Pm_{\text{variance}}$ refers to the preset probability variation. The sigmoid function is used to normalize the values. $V_i$ refers to the fitness function value of the individual of the current population, and $V_{i-1}$ refers to the fitness function value of the individual of the previous generation.

2.3.4. The Improved Genetic Algorithm Is Applied in Fault Localization

First, we treat the test cases as individuals. All test cases constitute the population, and the test cases are binary-coded to complete the population initialization. Then, in the calculation fitness phase, we calculate the fitness for each individual according to the fitness function. The higher the fitness level, the higher the quality of the test case, in order to improve the fault location effect, we propose the corresponding fitness function. Next, entering the selection phase, we select the test cases with high fitness according to a certain proportion as the parent generation of the next generation of test cases. In the crossover phase, we cross the selected parent test cases in pairs to generate the next generation of test cases. Mutate child test cases with a small probability during the mutation phase are used to enrich test cases. Finally, we determine whether the end test case generation condition is met; otherwise, we repeat the above process.

## 3. Experiment

### 3.1. Research Question

We conducted a series of experiments to evaluate the performance of FLFR. The experiments were designed with the following three questions:

- [RQ1] How efficient is the utilization of FLFR for different fault types?
- [RQ2] Can the improved genetic algorithm be used to generate effective test cases for software fault localization?
- [RQ3] How to determine the parameters used in the improved genetic algorithm?

### 3.2. Subject

To answer these questions, we selected four C programs from the Siemens suite in the SIR library (http://sir.csc.ncsu.edu/portal/index.php, accessed on 1 January 2023), one java program from Defect4j (https://github.com/rjust/defects4j, accessed on 1 January 2023) and one java program about elevator schedules. The information on these programs is shown in Table 3. The Siemens suite is one of the most commonly used evaluation programs in the field of software testing. Tcas is part of the aircraft collision avoidance system used to achieve aircraft altitude leveling; Schedule and schedule2 are the priority schedulers; tot_info generates statistics for the specified input data; print_tokens and print_tokens2 are lexical analyzers. For each program in the Siemens suite, there are a fault-free version and several faulty versions. Each faulty version contains one or more manually implanted faults. In addition, we also chose two Java programs—Commons-math and Elevator—to demonstrate the cross-language capabilities of the method proposed in this paper.

**Table 3.** Experimental programs.

| Program | Description | Language |
|---|---|---|
| Print_token | Lexical analyzer | C |
| Print_token2 | Lexical analyzer | C |
| Tot_info | Information measure | C |
| Tcas | Altitude separation | C |
| Commons-math | Java math library | Java |
| Elevator | Elevator scheduler | Java |

*3.3. Evaluation Criteria*

To evaluate the effectiveness of our approach, we adopted three metrics, which are EXAM score, NDCG (Normalize Discounted Cumulative Gain) [15], and EIO (the effect of the improved method relative to the original method).

EXAM is an evaluation index widely used in software fault localization. As shown in Equation (4), it is defined as the number of statements to check the program to find a fault_stat in total statements all_stat, The lower the stat ratio, the better the effect of fault localization.

$$EXAM = \frac{check\_stat}{all\_stat} \tag{4}$$

NDCG is a common index for evaluating information retrieval, which considers the recommendation order and the effective results contained in the recommendation. It can be used to evaluate the recommendation sequence in the retrieval of information. The output of software fault localization can be regarded as the recommendation of suspicious fault lines.

$$DCG(D, t, g, r) = \sum_{d \in D} \frac{g(d, t)}{s(r(d))} \tag{5}$$

$$NDCG(D, t, g, r) = \frac{DCG(D, t, g, r)}{\max_{l \in Dn} DCG(D, t, g, l)} \tag{6}$$

where $D$ is a group of documents, $t$ is a subject, $R(d)$ obtains the ranking of document $D$, and $G$ is the income function, which depends on the real score of subject $t$ predicted from document $D$, and $S$ is the distance function used for ranking. The score range is [0,1]. The higher the value, the better the effect.

The *EIO* index is defined in this paper to evaluate the effect of the improved method compared with the original method. A positive score represents the improvement of the effectiveness. A score of 0 represents no change in the effect, and a negative score represents a decrease of effectiveness.

$$EIO = \frac{org - evo}{org} \times 100\% \tag{7}$$

where *org* represents the number of lines of the original method to find a fault check and *EVO* represents the number of lines of the improved method to find a fault check.

*3.4. Variables*

3.4.1. Independent Variables–Methods under Comparison

For RQ1, we analyzed the fault from the source code and divided the fault types into three categories: missing expression statement (MIES), modified expression statement (MOES), and modified control condition (MOCC). Among them, the lack of expression statement includes the lack of assignment operation and arithmetic operation statement. Expression statement modification includes modification of operator or variable symbols in assignment operation, arithmetic operation statement, and modification of control conditions. Compared with 13 typical fault localization methods, namely *Kulcyznskil*,

*Tarantula, Jaccard, Ochiai, Ochiai2, $D^2Star$, $D^3Star$, Op2, Gp02, Gp19, BPNN, RBFNN, CNNFL.* The evaluation index uses the iterative optimization percentage EOP.

For RQ2, FLFR is compared with the traditional genetic algorithm (GA), and the random test case generation algorithm (RT). We conducted two experiments, namely the comparison experiment of FLFR with GA and RT, and the effect experiment of FLFR on different fault types.

In the comparison experiment of FLFR with GA and RT, we select FLFR, genetic algorithm generation test case method GA (Genetic Algorithm), and random test case generation method Random Testing. They are compared with the fault localization results generated without test cases. The experiment is based on three typical SBFL methods, Ochiai, Ochiai2, and D3̂Star, and the iterative optimization percentage *EIO* indicator is used to evaluate the results.

For RQ3, we conducted two experiments, namely, the iteration number experiment of generating test cases using FLFR, and the influence experiment of the objective function adjustment of the genetic algorithm on the results.

In the influence experiment of the objective function adjustment of the genetic algorithm on the results, we conducted this experiment to investigate whether the improvement of the fitness function of the key statement coverage has an impact on the fault localization effect. This experiment is for three coverages, named only feature importance coverage fit_i, line coverage fit_l, and branch coverage fit_b. We selected three typical spectral statistics methods Ochiai, Ochiai2, D3̂Star, and a typical deep learning method CNNFL. In addition, we record the results of the fault localization iteration process 10 times to compare the influence of the fitness function adjustment on the fault localization effect. We evaluated the research question with EXAM and NDCG metrics.

### 3.4.2. Parameter Settings

We used GCC and Jdk to compile the C programs and Java programs, respectively. For C programs, we applied *gcov* to obtain coverage information. For the Java program, we use the coverage tool Clover 4.4.1 to collect the spectrum information.

The parameters in the genetic algorithm are set as follows: the population size is set to 30, the maximum evolutionary generation is 20, the crossover rate is 0.4, and the mutation rate is 0.001. These parameter values are the results obtained after many experiments.

## 4. Experiment Result

### 4.1. Answer to RQ1—Performance on Different Failure Types

The experimental results are shown in Table 4. We can see that the control condition modification fault type is better than the other types in the eight fault localization methods, and the average index is 23.72%. The fault type modified by the expression statement is better than the other types among the three fault localization methods, and the average index value is 21.05%. The fault type of missing expression statement is better than other types only in two fault localization methods, and the average value of the index is 15.85.

We analyze the result and find that even if more test cases are generated, the missing fault statement will not be covered due to the missing type of expression statement. Therefore, this method has the least improvement in iterative fault localization. However, both the control condition modification fault type and the expression statement modification fault type will improve the positioning effect because there are more fault-related test cases.

We can conclude that the method in this paper improves the fault localization effect of two types of expression statement modification and control condition modification. However, the fault localization for the type of missing expression statement needs to be improved.

**Table 4.** Performance on different failure types.

| Subject | MIES (%) | MOES (%) | MOCC (%) |
|---|---|---|---|
| Kulcyznskil | 0 | 15.2 | 18.2 |
| Tarantula | 0.2 | 9.5 | 17.0 |
| Jaccard | 11.0 | 16.4 | 16.5 |
| Ochiai | 1.3 | 6.2 | 11.8 |
| Ochiai2 | 10.5 | 15.1 | 14.9 |
| $D^2Star$ | 5.4 | 7.0 | 10.8 |
| $D^3Star$ | 5.4 | 6.9 | 8.5 |
| Op2 | 8.1 | 3.4 | 7.7 |
| Gp02 | 8.1 | 7.3 | 7.2 |
| Gp19 | 0.4 | 0.8 | 8.0 |
| BPNN | 51.1 | 65.5 | 61.1 |
| RBFNN | 41.0 | 45.5 | 61.5 |
| CNNFL | 63.6 | 74.8 | 65.2 |
| Average | 15.85 | 21.05 | 23.72 |

*4.2. Answer to RQ2—The Effective of FLFR*

The experimental results are shown in Table 5. We can see that the effect of FLFR on fault localization is the most obvious, followed by GA, and the effect of RT ranks third. In addition, we also found that except for the Tcas dataset, which did not improve in the three methods, the other datasets improved in the three methods. This is because the TCAS dataset has only 173 lines of code, the complexity of the program is low, even if more test cases are generated, and the spectral coverage ratio of the program lines will not cause changes, so the test case generation will not affect the fault localization results.

We analyzed the results of RT and GA. For the RT algorithm, because it uses a random strategy to generate test cases, there is no guarantee that the test cases will cover as many statements containing failures as possible. In contrast to RT, FLFR employs an overlay-based strategy that generates test cases that are helpful for fault location. Therefore, in terms of the effect of software fault location, FLFR is better than RT.

Compared with the genetic algorithm GA, FLFR not only generates test cases by simulating population evolution, but also improves individual generation strategies. Therefore, FLFR generates test cases through multiple iterations of software fault location results, so that the generated test cases achieve better results in software fault location than GA.

Therefore, we can conclude that the FLFR method improves the software fault localization effect.

**Table 5.** Compared to GA and RT.

| Subject | $Ochiai(\%)$ | | | $Ochiai2(\%)$ | | | $D^3Star(\%)$ | | |
|---|---|---|---|---|---|---|---|---|---|
| | FLFR | GA | RT | FLFR | GA | RT | FLFR | GA | RT |
| Print-token | 23.1 | 16.6 | 18.1 | 5.7 | 0 | 0 | 19.7 | 20.4 | 0 |
| Print-token2 | 38.7 | 12.3 | 0 | 5.7 | 3.70 | 1.8 | 6.2 | 8.8 | 2.9 |
| Tot-info | 22.4 | 19.6 | 0 | 10.5 | 7.92 | 1.1 | 8.4 | 3.10 | 1.25 |
| Tcas | 0 | 0 | 0 | 0 | 0 | 0 | 0 | 0 | 0 |
| Commons-math | 68.9 | 63.9 | 33.3 | 74.9 | 74.6 | 74.5 | 61.4 | 41.4 | 61.3 |
| Elevator | 82.0 | 38.4 | 7.37 | 73.2 | 19.7 | 0 | 51.8 | 11.4 | 0 |
| Average | 39.18 | 25.13 | 20.34 | 28.33 | 17.65 | 12.90 | 24.58 | 14.18 | 10.91 |

*4.3. Answer to RQ3—Ablation Studies*

4.3.1. Analysis of the Number of Iterations

To determine the number of iterations of FLFR, we conduct related experiments. The results of the experiment are shown in Figure 2, where the abscissa in the figure is the number of iterations, ranging from 1 to 10, and the ordinate is the evaluation index EXAM score. From the results, it can be seen that the images are divided into two types, where one is in the three datasets of Print-token, Elevator, and Tcas. After many iterations, the fault localization indicator tends to be stable; the other is in the three datasets of Print-token2, Tot-info, and Commons-math. There are peaks and valleys in the iterative process. The stable regions and peaks and valleys of these images represent the optimal results in the iterative process. It can be seen that except for the Tcas dataset, the fault localization effect has stabilized after 10 iterations.

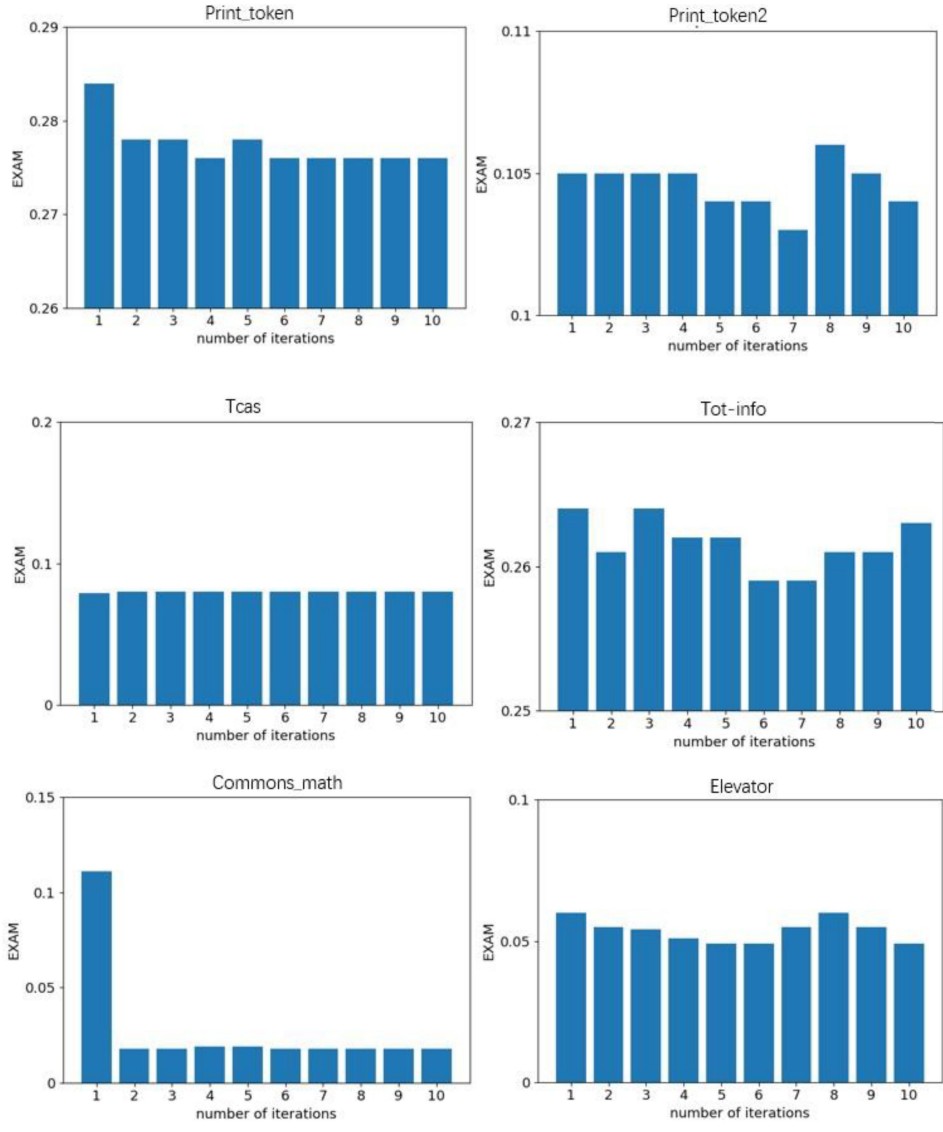

**Figure 2.** The number of iterations.

We analyzed the results. During the iterative localization process, the fault localization result of this round affects the generation of test cases, and the newly generated test cases are used as the input for the next round of fault localization, which also affects the result of fault localization. After many iterations, we found that more test cases covered the possible faulty program lines, the fault localization effect gradually reached the best, and

the image formed by the evaluation index was close to the optimal value, forming image stable areas and peaks and valleys. From the results of the 6 programs so far, 10 iterations is a good choice.

### 4.3.2. Adjustment of the Fitness Function

The experimental results are shown in Figures 3 and 4. We can see that, in addition to the CNNFL method in the adjustment of the three fitness functions, the positioning effect is close. Among other fault localization methods, the three genetic algorithm optimization functions only consider fit_i in most cases. The effect is better.

We have analyzed the results. For the CNNFL method, the scale of training data can affect its fault localization effect. However, the fitness function type does not cause changes in test case size. Therefore, we can see that the effect of the fault localization method for deep learning is not affected by the adjustment of the fitness function.

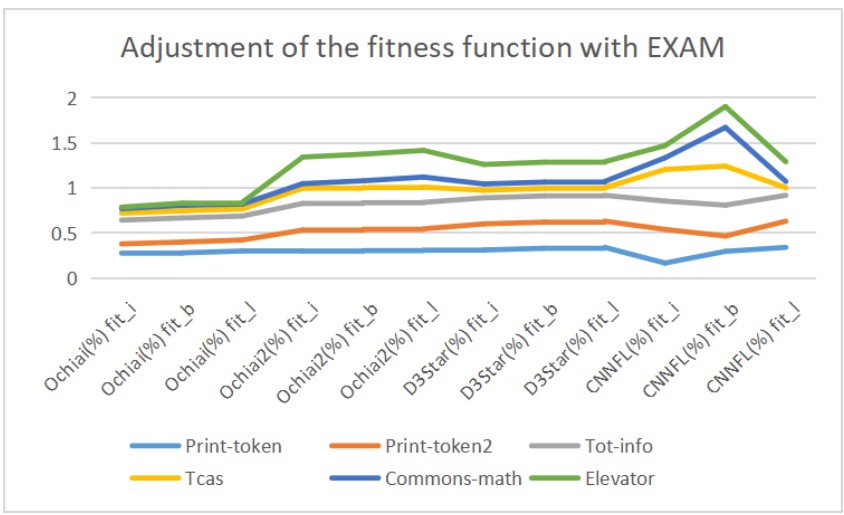

**Figure 3.** Adjustment of the fitness function with EXAM.

However, spectrum-based fault localization methods depend on the quality of the test cases. The fitness function adjustments of fit_l and fit_b both generate test cases that cover a wider range. However, the fitness function adjustment of fit_i can generate more test cases covering suspected fault lines, which is not sufficient for the statements that may generate faults.

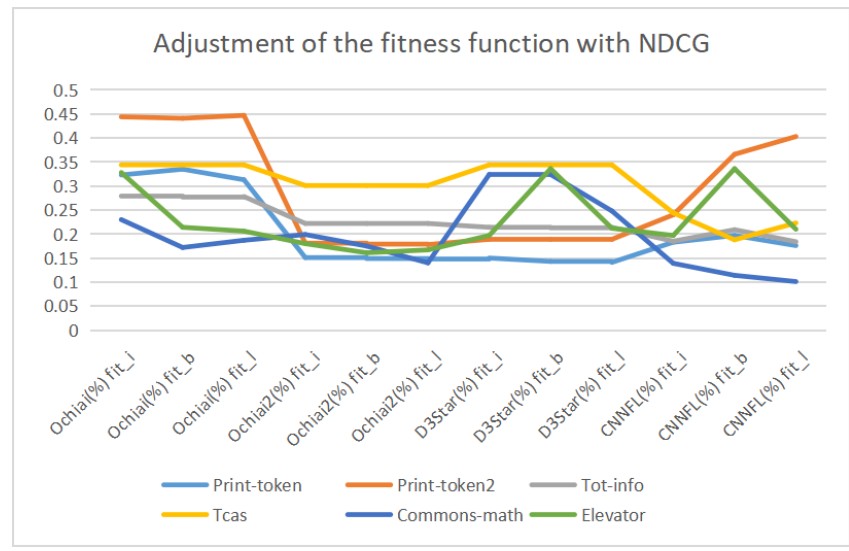

**Figure 4.** Adjustment of the fitness function with NDCG.

Therefore, we can conclude that the improvement of the fitness function in the genetic algorithm in this paper improves the software fault localization effect.

### 4.4. Discussion

The influence of the complexity of the program on the FLFR. We found that the complexity of the program plays an important role. In the experiment, we use the TCAS dataset, because the TCAS dataset only has 173 lines of code, the complexity of the program is low, even if more test cases are generated, and the spectrum coverage ratio of the program line will not change, resulting in the generation of test cases that will not affect the fault location effect. For programs with high program complexity, FLFR improves the quality of test case generation by simulating population evolution, improves the individual selection strategy, and makes the generated test cases more suitable for improving the fault location effect through multiple iterations of test case generation and software fault location results. These have also been verified experimentally.

The influence of error type on the FLFR. We also found that FLFR has better fault localization performance with two error types, which are expression statement modification and control condition modification. We analyze the types of faults from the source code and divide the fault types into three categories: missing expression statements, expression statement modification, and control condition modification. The performance of FLFR in these three types of errors is not completely consistent, for example, for the expression statement modification and control condition modification, the positioning effect of FLFR is significantly improved among the 13 fault location methods.

However, for the missing types of expression statements, the positioning effect of FLFR is only significantly improved in the three fault location methods. We analyze the reasons for this, and for the faults of the missing type of expression statements, even if more test cases are generated, the missing fault statements will not be covered, so the fault location effect of the iteration is not greatly improved. For the other two fault types, because more relevant test cases are generated, the positioning effect of multiple fault location methods is better.

The effect of the improvement of the fitness function of the genetic algorithm on FLFR. The efficiency of FLFR depends on the test case generation of the genetic algorithm, so we experimented with whether the improvement of the fitness function of the genetic algorithm had an impact on the fault localization effect. The experimental results show that when the fitness function is adjusted, more test cases covering the suspicious fault lines can be generated, and the fault location effect of FLFR will be better. If there are no more test cases covering suspicious fault lines during the adaptation function adjustment, the fault location effect of FLFR may not be obvious. Therefore, the improvement of the fitness function of the genetic algorithm needs to be closely used with the corresponding coverage criterion, so that FLFR can achieve better results.

Disadvantage and advantage of the FLFR. One disadvantage of FLFR is that it can be time-consuming and require significant manual effort, particularly if the system is large and complex. Additionally, the effectiveness of this approach depends heavily on the quality and accuracy of the feedback data being collected, which can sometimes be difficult to obtain or interpret correctly.

However, FLFR allows developers to quickly identify and isolate problems within the system, often resulting in faster resolution times than traditional methods. Additionally, because this approach relies on actual usage data rather than guesswork or speculation, it can often provide more accurate information about where the fault lies within the codebase.

### 4.5. Threats of Validity

By conducting our experiments, we found the following factors that might affect the validity of our study.

Implementation of baselines. The internal threat to validity is concerned with our implementation. We reproduced 13 typical fault localization methods, namely *Kulcyznskil*,

*Tarantula*, *Jaccard*, *Ochiai*, *Ochiai*2, $D^2Star$, $D^3Star$, *Op*2, *Gp*02, *Gp*19, *BPNN*, *RBFNN*, and *CNNFL*. Although we have implemented these baseline methods as described in the original studies, we cannot guarantee that these implementations exactly match the original ones.

Applying baselines on our dataset. In carrying out the task, we found that many of the baseline methods were designed specifically for a particular task, for example, *BPNN*, *RBFNN*, and *CNNFL* used the training data to obtain the model. Although we compared these methods as baselines, we cannot guarantee that we can meet the conditions for these representations of the model to work well.

## 5. Related Work

### 5.1. Spectrum-Based Fault Localization

Spectrum-based fault localization (SBFL) techniques are widely studied [16–21], which has been extended in various ways by considering inputs other than statement coverage [1], control flow analysis [2], and data flow analysis [3].

For example, Jones et al. [1] used the statement coverage information, which applied color to visually map the participation of each program statement in the outcome of the execution of the program with a test suite. This technique recorded the statement coverage status of the test case execution and whether the test case is executed successfully. For each statement in the code, the execution status of the failed test case and the pass test case can help the user locate potentially faulty statements. Abreu et al. [14] investigate this diagnostic accuracy as a function of several parameters (such as the quality and quantity of the program spectra collected during the execution of the system). They used the similarity coefficients Ochiai in the field of molecular biology to define the probability of failure of statements in a program. Wong et al. [4] obtained the coverage information of each executable statement and the execution result (pass or fail) with respect to each test case by using a crosstab-based statistical method.

In addition, Renieris et al. [22] compared the correct runs and failed runs. They assumed the existence of a faulty run and a larger number of correct runs and adopted a technique that compares the spectra corresponding to correct run and faulty run. Then, they produced a report of suspicious parts of the program. Laleh et al. [5] presented a technique that obtained a ranking of statements in terms of their likelihood of being faulty by leveraging the result of combinatorial testing. Wen et al. [23] proposed a technique, namely HSFL (spectrum-based fault localization), to leverage the information of version histories in fault localization.

### 5.2. Program State Analysis Based Fault Localization

There exist fault localization techniques that use program state analysis. For example, Zeller et al. [24] applied Delta Debugging multiple states of the program, which automatically reveals the cause-effect chain of the failure. Masri et al. [25] proposed a fault localization technique based on dynamic information flow analysis, which considers dynamic direct control dependencies, dynamic direct data dependencies, and other dynamic dependencies between programs to locate faults. Zhang et al. [26] used the edge profiles to represent passed executions and failed executions. They analyzed the propagating infected program states for fault localization.

In addition, some researchers abstract the program into a graph structure and use the graph structure to locate software faults. For example, Baah et al. [27] exploited an extended program dependency graph for fault localization. The method is based on a probabilistic graphical model. First, they obtain the program dependency graph through the execution information of the test cases. Then, based on the program dependency graph, a probabilistic program dependence graph (PPDG) is established by adding nodes and edges. Next, they calculated the conditional probabilities of all nodes that the program would execute if one failed, using the relationship of uncertainty that exists in the probability graph. When

the conditional probability of a node is smaller, it means that the node's fault probability is higher.

### 5.3. Machine Learning Based Fault Localization

There exist many machine learning-based fault localization methods. For example, Wong et al. [28] proposed a fault localization method based on the Back Propagation Neural Network (BPNN), which is first trained according to the execution information of each test case. Then, they used dummy test case vectors as input to the test set, where each dummy test case vector covered only one line of statements. Finally, the network outputs the degree of suspicion that each line of the statement is faulty. In addition, Wong et al. [29] used an improved radial basis function neural network RBFNN (Radial Basis Function Neural Network) for software fault localization. They propose a weighted dissimilarity function to estimate the distance between the sentence coverage vectors of two test cases, which solves the problem of the BP network getting stuck in a locally optimal solution.

In addition, there are a few software fault localization studies with deep learning techniques. For example, Zhang et al. [8] proposed a convolutional neural network-based fault localization method CNNFL (Convolutional Neural Network Fault localization). Maru et al. [30] proposed an effective approach for fault localization based on a back-propagation neural network that utilizes branch and function coverage information along with test case execution results to train the network. Dutta et al. [31] combined spectrum localization and neural network technology, and proposed a hybrid fault localization approach. Li et al. [32] proposed a deep learning approach, namely DeepFL, to automatically learn the most effective existing/latent features for precise fault localization.

### 5.4. Fault Localization in Deep Neural Networks

There may also be faults in deep neural networks, so some researchers have carried out fault location research on deep neural networks. For instance, Meng et al. [33] proposed an approach, namely TRANSFER, which leverages the deep semantic features and transferred knowledge from open-source data to improve fault localization. Wardat et al. [34] proposed a debugging approach, namely DeepDiagnosis, which localizes the faults for DNN programs. Eniser et al. [35] applied spectrum-based fault localization techniques to systematically identify suspicious neurons and then uses these neurons to synthesize new inputs, which is used for DNN testing and verification. It counts the number of times a neuron was active/inactive when the network made a successful or failed decision. It then calculates a suspiciousness score such as the spectrum-based fault localization tool Tarantula.

### 5.5. Test Case Generation

The manual software test case generation relies on the experience and level of the testers to a large extent, which is time-consuming. To this end, many researchers have proposed a large number of automated test case generation methods, which include search-based test case generation [36–39], constraint solving-based test case generation [40–42], requirement-based test case generation [43,44], symbol-based test case generation, executed test case generation [45–48], and random test case generation method [49]. Search-based test case generation originates from research in artificial intelligence. To solve some problems that are difficult to solve accurately, we often convert it into a process of searching in the problem space. The search-based test case generation method generates the next search action according to the information obtained from the previous search results. Search algorithms are an efficient way to deal with NP-hard problems. Commonly used search algorithms include hill climbing [50], simulated annealing algorithm [51], and genetic algorithm [52].

The test case generation approach based on symbolic execution uses symbols instead of actual values in the process of substituting into the program for operation. Thus, the constraints and calculation formulas of specific paths can be analyzed [45]. There are two

main ways to build a restraint system. One is to replace from front to back, and the other is to replace from back to front. The random test case generation method is simple and effective, which can obtain a relatively high code coverage rate by using a small number of test cases.

All these techniques have achieved good results, which are the basis of our research.

## 6. Conclusions

In this paper, a software fault localization method based on an improved genetic algorithm, namely FLFR , is proposed. First of all, we describe the existing fault localization methods. Then, we find the quality of the used test cases that affect the suspiciousness ranking and thus the efficacy of fault localization. As such, it is necessary to generate test cases with "better" quality to improve the chance of faulty statements being ranked as highly suspicious. At last, in view of the existing problems, the research content of this paper is proposed.

FLFR uses the existing fault localization method to calculate the program fault suspiciousness score ranking and then generates test cases according to the ranking information. To improve the quality of test cases, the final ranking of fault suspicion is displayed after many iterations, so as to improve the effect and efficiency of software fault localization. We compared the typical 13 fault localization and 2 test case generation methods with 6 datasets. The experimental results show the effectiveness of our proposed software fault localization effect.

However, the software fault localization method in this paper has a low effect on improving the faults of missing statement types because of fault localization based on spectrum statistics, as the missing statement types cannot be covered in the program spectrum. Second, when performing iterative fault localization, we still empirically choose the number of iterations based on experimental data. In addition, we will explore different optimization techniques, or integrate machine learning methods for even more accurate fault localization. Finally, our method has not been applied in more practical cases, which is also the direction of our follow-up research.

**Author Contributions:** Formulation or evolution of overarching research goals and aims. Management and coordination responsibility for the research activity planning and execution, B.Y.; Application of statistical, mathematical, computational, or other formal techniques to analyze or synthesize study data, X.M.; Application of statistical, mathematical, computational, or other formal techniques to analyze or synthesize study data, H.G.; Application of statistical, mathematical, computational, or other formal techniques to analyze or synthesize study data, Y.H.; Management and coordination responsibility for the research activity planning and execution. Oversight and leadership responsibility for the research activity planning and execution, including mentorship external to the core team. Provision of study materials, reagents, materials, patients, laboratory samples, animals, instrumentation, computing resources, or other analysis tools, F.X. All authors have read and agreed to the published version of the manuscript.

**Funding:** This work was supported by National Key R&D Program of China (2022YFF1302700), The Emergency Open Competition Project of National Forestry and Grassland Administration (202303), Outstanding Youth Team Project of Central Universities (QNTD202308).

**Institutional Review Board Statement:** Not applicable.

**Informed Consent Statement:** Not applicable.

**Data Availability Statement:** Not applicable.

**Conflicts of Interest:** The authors declare no conflict of interest.

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
