# Peer review of "An Approach of Improving the Efficiency of Software Fault Localization based on Feedback Ranking Information"

_applsci, doi:10.3390/app131810351_

Round 1

Reviewer 1 Report

There are some comments that you could address:

  1. Clarify the Contribution: While the paper presents an innovative approach, it would be beneficial to explicitly outline the unique contributions and advantages of the FLFR method in the introductory sections. This will help readers quickly grasp the novel aspects of your research.
  2. Elaborate on Methodology: Provide a more detailed explanation of how the improved genetic algorithm is applied in the FLFR approach. Describe the specific parameters or modifications made to enhance the algorithm's performance and highlight any comparisons with traditional genetic algorithms.
  3. Address Potential Limitations: Discuss potential limitations of the FLFR approach, such as scenarios where it might not perform optimally. This would give readers a more comprehensive understanding of the method's applicability.
  4. Real-World Applicability: Illustrate the practical scenarios where FLFR could have a significant impact. Are there specific industries or domains where accurate fault localization is crucial? This could enhance the relevance and potential impact of your research.
  5. Benchmarking Metrics: Consider incorporating additional benchmarking metrics beyond just "effectiveness." Metrics like execution time, resource utilization, or scalability could provide a more holistic evaluation of FLFR's performance.
  6. Case Study or Example: Integrate a real-world case study or example to demonstrate the step-by-step implementation of FLFR. This could help readers visualize how the approach works in a practical setting.
  7. Sensitivity Analysis: Perform a sensitivity analysis to evaluate the robustness of FLFR against variations in input parameters or different dataset characteristics. This would provide insights into the stability and reliability of your approach.
  8. Comparison with State-of-the-Art: Extend the comparison section to include a detailed analysis of FLFR against state-of-the-art fault localization methods. Highlight not only its superiority but also where it might fall short and potential areas for improvement.
  9. Visualization: Incorporate visual aids, such as graphs or charts, to visually represent the improvement in fault suspicion ranking achieved by FLFR over iterations. Visualizations can enhance the clarity of your results.
  10. Impact on Debugging Process: Discuss the broader implications of FLFR on the software debugging process as a whole. How might adopting this approach lead to faster bug identification and resolution, ultimately improving software development efficiency?
  11. User-Friendly Interface: Consider discussing the user interface or platform through which FLFR can be accessed and utilized. A user-friendly interface could encourage wider adoption among software developers.
  12. Future Work: Conclude the paper by outlining potential directions for future research. This could include refining the FLFR algorithm, exploring different optimization techniques, or integrating machine learning methods for even more accurate fault localization.
  13. Code Availability: If feasible, provide access to the source code or a simplified implementation of FLFR. This would enable other researchers to replicate your experiments and further validate your findings.
  14. Citation of Related Works: Ensure that relevant recent works in the field of fault localization and genetic algorithms are cited and discussed. This strengthens the context and background of your research.
  15. Peer Review: Consider seeking feedback from colleagues or experts in the field before finalizing your paper. Peer review can provide valuable insights and suggestions for further improvement.
  16. Finally, I wish authors could do a brief bibliometric analysis, such as referring to -Fault tree analysis improvements: A bibliometric analysis and literature review
  17. And discuss about uncertainty handling, such as referring to: Uncertainty modeling in risk assessment of digitalized process systems

NA

Author Response

Thank you for your email on Aug 17, 2023 regarding our paper entitled “An Approach of Improving the Efficiency of Software Fault Localization based on Feedback Ranking Information” submitted to Applied Sciences (Manuscript ID: applsci-2567195).

We have revised our paper according to the comments from the editor and reviewers. We are submitting a revised version. Please refer also to the appendix of this cover letter for detailed responses the comments.

Reviewer 2 Report

The author proposed FLFR approach for software fault localization. The fundamental concept behind FLFR is the repeated refinement of test cases based on feedback ranking data gathered from prior fault localisation outcomes. This method is suggested in the study as a solution to the problem of low-ranked real defective statements in the context of fault localization.

 Strengths of the paper:

1.       The FLFR technique offers a distinctive viewpoint by providing feedback from earlier attempts at fault localization into the process of creating test cases. This iterative process may produce better fault localisation outcomes.

2.       The framework provided in the paper was concise and clear for the proposed FLFR approach. Also steps involved in test case creation and execution, iterative optimization and fitness function enhancements are well-defined.

3.       In result the FLFR evaluation was done using various metrics and experimental datasets.

Weaknesses of the paper:

1.       I found multiple repeated statements throughout the paper. A few examples are:

The framework of the method proposed in this paper ---------------------- In the framework of the method proposed in this paper.

We can conclude that the method in this paper…. This statement is repeated multiple times it could have been explained in a single statement for all methods in the end.

2.       The method is intriguing, but the paper does not provide a thorough theoretical justification for why the FLFR iterative procedure should produce improved fault localisation results. More information about the underlying factors causing this improvement could be provided by the authors.

3.       The FLFR approach's potential limits must be discussed in the paper in order to understand how well it can be applied and where it can fall short.

4.       And finally the FLFR method was implemented but empirically choosing iterations. While it can be a good parameter for testing the algorithm the implications of using iterations for justifying the algorithm is still questionable. It would have been very interesting to see fault localizations on practical real life scenarios.

Please also check on the references. Some references are missing, as [?] is available at three places in the text.

Author Response

(The authors gave the same response as above.)

Round 2

Reviewer 1 Report

Well done

NA

Author Response

Dear Prof. Astrid,

An Approach of Improving the Efficiency of Software Fault Localization

based on Feedback Ranking Information

by Bo Yang, Haoran Guo, and Yuze He

(Manuscript ID: applsci-2567195)

Thank you for your email on Aug 28, 2023 regarding our paper entitled “An Approach of Improving the Efficiency of Software Fault Localization based on Feedback Ranking Information” submitted to Applied Sciences (Manuscript ID: applsci-2567195).

We have revised our paper according to the comments from the editor and reviewers. We are submitting a revised version. Please refer also to the appendix of this cover letter for detailed responses the comments.

We look forward to hearing from you.

Sincerely,

Bo Yang

Enclosure

Reviewer 2 Report

Minor correction:

1. Add a reference to the existing methods in the introduction.

2. Figure 4 and Table 2 are never referred.

Author Response

(The authors gave the same response as above.)

Round 3

Reviewer 1 Report

Well done.